# C-Reactive Protein as a Predictor of Survival and Length of Hospital Stay in Community-Acquired Pneumonia

**DOI:** 10.3390/jpm12101710

**Published:** 2022-10-13

**Authors:** Apostolos Travlos, Agamemnon Bakakos, Konstantinos F. Vlachos, Nikoletta Rovina, Nikolaos Koulouris, Petros Bakakos

**Affiliations:** 1st University Department of Respiratory Medicine, National and Kapodistrian University of Athens, 11527 Athens, Greece

**Keywords:** C-reactive protein (CRP), community-acquired pneumonia (CAP), biomarkers, mortality, prognosis, length of stay

## Abstract

Introduction: Community-acquired pneumonia (CAP) presents high mortality rates and high healthcare costs worldwide. C-reactive protein (CRP) has been widely used as a biomarker for the management of CAP. We evaluated the performance of CRP threshold values and ΔCRP as predictors of CAP survival and length of hospital stay. Methods: A total of 173 adult patients with CAP were followed for up to 30 days. We measured serum CRP levels on days 1, 4, and 7 (D1, D4, and D7) of hospitalization, and their variations between different days were calculated (ΔCRP). A multivariate logistic regression model was created with CAP 30-day survival and length of hospital stay as dependent variables, and absolute CRP values and ΔCRP, age, sex, smoking habit (pack-years), pO2/FiO2 ratio on D1, WBC on D1, and CURB-65 score as independent variables. Results: A total of six patients with CAP died (30-day mortality 3.47%). No difference was found in CRP levels and ΔCRP between survivors and non-survivors. Using a cut-off level of 9 mg/dL, the AUC (95% CI) for the prediction of survival of CRP on D4 and D7 were 0.765 (0.538–0.992) and 0.784 (0.580–0.989), respectively. A correlation between CRP values on any day and length of hospital stay was found, with it being stronger for CRPD4 and CRPD7 (*p* < 0.0001 and *p* = 0.0024, respectively). A reduction of CRP > 50% from D1 to D4 was associated with 4.11 fewer days of hospitalization (*p* = 0.0308). Conclusions: CRP levels on D4 and D7, but not ΔCRP, could fairly predict CAP survival. A reduction of CRP > 50% by the fourth day of hospitalization could predict a shorter hospital stay.

## 1. Introduction

Community-acquired pneumonia (CAP) is a major public health problem with a high mortality rate of approximately 5–15% and a considerable socio-economic burden worldwide [1,2]. It remains a main reason of hospitalization, death, and high healthcare costs in developed countries, especially among elderly people [3,4].

CRP is an acute-phase protein predominantly produced in the liver. Responding to infection or tissue inflammation, the production of CRP is rapidly stimulated by cytokines, particularly interleukin (IL)-6 [5]. Moreover, CRP is not only a marker, but also a driver of inflammation by human macrophages [6,7].

In patients with severe CAP, a decrease of less than 25% in CRP levels at the second day was significantly associated with 30-day all-cause mortality [8]. Additionally, a failure of CRP to fall by 50% or more at day 4 leads to an increased risk for 30-day mortality and a need for mechanical ventilation [9]. Conversely, a recent 5-year follow-up cohort study showed no higher admission levels of CRP in patients with CAP experiencing an adverse short-term outcome (intensive care unit admission and 30-day mortality) [10]. CRP levels have been shown to decrease after successful antibiotic treatment, and serial assessment of CRP aided in the early identification of CAP patients with poor outcomes [11,12].

The correlation of CRP values with the length of hospital stay has also been evaluated. In 823 adult patients hospitalized with CAP, a lack of a CRP decline within three days of hospitalization was associated with a high risk of complications and a prolonged hospital stay [13]. Similarly, in a study from Sweden, it was demonstrated that hospital-treated CAP patients with high IL-6 or CRP levels had a longer duration of fever and a longer hospital stay [14]. A reduced length of hospital stay can result in substantially lower costs [15].

The aim of the present study was to assess the predictive value of CRP levels in patients with CAP. We hypothesized that serum CRP levels on days 1 (D1), 4 (D4), and 7 (D7) could predict survival and hospital length of stay. We carried out a study in which we evaluated: (1) serum CRP levels on the first, fourth, and seventh day of CAP; and (2) ΔCRP, as a prognostic marker of CAP survival (up to D30) and length of hospital stay. We also evaluated whether CRP in any day measured was associated with the severity of respiratory failure.

## 2. Materials and Methods

### 2.1. Study Design and Population

The study was conducted at the 1st University Department of Respiratory Medicine, University of Athens, from January 2013 until September 2019. The local ethics committee approved the study.

Initially, 194 patients with CAP were screened. Patients with hospital-acquired pneumonia (HAP), immunocompromised patients (hematologic malignancies, HIV, neutropenia < 1000 cells/mL, and patients who had received chemotherapy or other immunosuppressive therapy over the past 2 months) were excluded from the study. Patients who died within the first 2 days of CAP diagnosis were also excluded from the study. Moreover, patients who had received antibiotic treatment at least 1 day prior to hospital admission were not included in the study. Finally, 173 patients with CAP comprised the study group. The day of CAP diagnosis was defined as D1 and was the day that empirical antibiotic treatment was started. The following days were termed as D2, D3, etc. CRP levels were measured on D1, D4, and D7 in all patients included in the study. Patients were followed until the 30th day after CAP diagnosis and then were considered survivors. Those who died before D30 were considered non-survivors. Antibiotic treatment was chosen by the treating physician and was modified according to the susceptibility pattern of the sputum and/or blood culture in case empirical treatment did not cover the isolated pathogen. Blood samples were collected on D1, D4, and D7. The samples were centrifuged at 2500 rpm for 15 min, and the plasma was aliquoted and stored at −80 °C until analyzed in a single batch. Circulating levels of CRP were measured using an immunoturbumetric method with a commercially available kit (Dade Behring). Normal values for CRP were <0.70 mg/dL.

### 2.2. Statistical Analysis

Values were expressed as mean (±SD) or median (interquartile range 25–75 percentile) in the case of a skewed distribution. Comparisons between patient groups were performed by using the Mann–Whitney U-test method. The difference Δ was calculated using the formula: Δ = D4-D1, D7-D4 and D7-D1, respectively. Therefore, ΔCRP4-1 = CRPD4-CRPD1, ΔCRP7-4 = CRPD7-CRPD4, and ΔCRP7-1 = CRPD7-CRPD1, where Δ > 0 refers to increasing values and Δ ≤ 0 refers to reducing values. The ΔCRP values were classified as increasing or unchanging/decreasing. A univariate logistic regression analysis followed in order to define the risk factors for CAP survival and hospital length of stay. A multivariate logistic regression analysis model was created with CAP survival and hospital stay as dependent variables, whereas the absolute CRP values on D1, D4, and D7, as well as the changes in CRP values on days D1, D4, and D7 (ΔCRP4-1, ΔCRP7-1, and ΔCRP7-4), were set as the as independent variables. In order to deal with possible linearity, models were created that contained only absolute values or only changes, as well as models with absolute values and changes combined together. In order to control for potential confounding factors, age, gender, and CURB-65 score were included in the original model. Results were reported as ORs, adjusted at 95% CI.

Sensitivity and specificity were calculated. Threshold values that gave the best combination of sensitivity and specificity were judged by calculating the Youden’s index, i.e., the maximum difference between sensitivity and specificity [16].

The SPSS statistical package was used. A *p*-value < 0.05 was considered significant.

## 3. Results

A total of 173 patients with CAP were eventually included in the study. All of them were hospitalized in the 1st University Department of Respiratory Medicine, University of Athens, Greece. A flowchart of the study population is shown in Figure 1. The demographic characteristics and the CRP values are shown in Table 1.

In 11 patients, a positive sputum culture was found (Klebsiella Pneumoniae: 3, Staphylococcus Aureus: 1, Stenotrophomonas Maltophila: 1, MRSA: 1, Pseudomonas Aeruginosa: 3, other Gram (+) bacteria: 1, and other Gram (−) bacteria: 1). A total of 4 patients had a positive blood culture (Klebsiella Pneumoniae: 1, Streptococcus Pneumoniae: 1, Proteus Mirabilis: 1, and other Gram (+) bacteria: 1), while in 4 patients with CAP, a positive urine antigen for strep pneumonia (*n* = 3) and for Legionella (*n* = 1) was detected.

### 3.1. CAP Survival

CRP values exceeded normal levels in 172 out of 173 patients on D1.

During the study period, six patients with CAP died (3.47%). One death occurred on D3 of hospitalization, and five non-survivors died between D8 and D23. Non-survivors were older compared to survivors (*p* = 0.011). There was no difference in CRP levels between survivors and non-survivors on any day although non-survivors had higher CRP levels, especially on D4 and D7.

In the univariate analysis, CRPD4 and CRPD7 were able to predict survival of CAP. However, this predictive performance was lost in the multivariate analysis.

ΔCRP scores were not different between survivors and non-survivors either.

Using as cut-off level the value of 9 mg/dL, the AUC and 95% CI for the prediction of survival for CRP on D4 and D7 were 0.765 (0.538–0.992) and 0.784 (0.580–0.989), respectively (Figure 2A,B).

Moreover, neither a reduction of CRP >50% from D1 to D4, nor a reduction >50% from D4 to D7 were associated with better survival (*p* = 0.0688 and 0.362, respectively).

CURB-65 was not associated with mortality (*p* = 0.512), and the addition of CURB-65 to absolute CRP values (D1, D4, and D7) did not improve its performance.

Using a cut-off value of 3 for CURB-65 (thus <3 and ≥3), the AUC and 95% CI for the prediction of survival was 0.702 (0.5245–0.8801).

One patient with a positive blood culture died, while no pathogen was isolated in the cultures of the other five non-survivors. Compared to those with a negative blood culture, patients with a positive blood culture had significantly higher CRP levels on D1 (*p* = 0.016), while levels on D4 and D7, although higher, did not reach significance (*p* = 0.128 and *p* = 0.077, respectively).

### 3.2. CAP Hospital Length of Stay

A correlation between CRP values on any day and length of hospital stay was found, being stronger for CRPD4 and CRPD7 (*p* < 0.0001 and *p* = 0.0024, respectively).

Furthermore, a reduction of CRP >50% from D1 to D4 was associated with a shorter hospital length of stay and corresponded to 4.11 fewer days of hospitalization (*p* = 0.0308). A reduction of CRP >50% from D4 to D7 corresponded to 1 fewer hospitalization days, which was not significant (*p* = 0.5657).

The CURB-65 score was not associated with hospital length of stay (*p* = 0.2762).

Additionally, a positive correlation was detected between PaO2/FiO2 and CRPD4 or CRPD7 (*p* = 0.0008 and *p* = 0.0392, respectively).

## 4. Discussion

Several biomarkers have been evaluated in an effort to assess the prognosis of patients with CAP, and these measurements are supplementary to traditional clinical tools.

CRP has been established as a widely used inflammatory biomarker in CAP and has been evaluated in several studies. It is included in the clinical protocols for CAP of several hospitals [17], and it is mentioned in the guidelines of lower respiratory tract infections (LRTIs) [18]. Currently, CRP and PCT are the best available tools to assess the severity of CAP [18]. When CRP levels remain unremarkable or low at follow-up measurements, a relevant severe infection is very unlikely [19].

In the present study, we aimed to evaluate the predictive performance of CRP in survival and hospital length of stay in hospitalized patients with CAP. The choice of D1, D4, and D7 was arbitrary and was based upon the clinical course of CAP, thus at admittance (D1), after three days of antibiotic treatment when re-evaluation usually occurs (D4), and towards the end of antibiotic treatment (D7). We did not find any association between absolute CRP values measured on D1, D4, and D7 and survival of patients hospitalized with CAP. Neither did we find any association between ΔCRP and CAP survival. However, a failure to reduce CRP levels <9 mg/dL on D4 and D7 after the initiation of antibiotic treatment could fairly predict CAP survival. Moreover, we found that a reduction of CRP of more than 50% from D1 to D4 could predict a shorter hospital length of stay. The findings of our study regarding mortality are in contrast to the findings of other studies. A retrospective study from Denmark, including 814 patients with CAP, showed that absolute CRP levels and relative decline on the third day of hospitalization were both predictors of 30-day mortality. Moreover, the highest mortality risk was found in CAP patients with a level of CRP > 75 mg/L who failed to decline 50% by day 3 [20]. Another study demonstrated that a failure to present a decline in CRP levels was associated with a poor prognosis, irrespective of the actual level of CRP [21]. Similarly, according to the German competence network CAPNETZ, in CAP patients without antimicrobial pre-treatment, survivors had lower values of CRP, as well as PCT and WBC, compared to non-survivors, and these biomarkers predicted 28-day mortality exclusively in these patients. However, in patients with antimicrobial pre-treatment, the values of PCT, WBC, and CRP did not differ significantly in survivors and non-survivors, indicating that there is an effect of antibiotic pre-treatment in the levels of inflammatory biomarkers [22]. This discrepancy may be attributed to the low number of deaths in our study population. Only 6 out of 173 patients died, and this corresponds to a percentage of 3.47%, which is lower in comparison to other studies. Moreover, the mean age of survivors was lower in comparison to other studies, possibly contributing to the low mortality since mortality has been shown to increase with age [23].

Patients with a positive blood culture had higher CRP levels on D1, indicating that they constituted a more severely ill population. However, this difference was not observed in D4 and D7 CRP levels, and most importantly, it did not influence the outcome (death or survival), demonstrating a less crucial effect after the initiation of antibiotic therapy.

The finding of an association of ΔCRP with hospital length of stay is important. CAP presents a varying spectrum of severity, and hospital stay increases its cost and morbidity significantly. We found that, of those patients with CAP in whom CRP decreased from D1 to D4, more than 50% stayed in the hospital for four fewer days. The decision of hospital discharge was based upon the clinician’s decision and other factors, such as social factors and comorbidities. However, apart from being statistically significant, four fewer days of hospital stay are clinically important as they correspond to 30% less hospitalization time and a much lower cost. A relatively small decrease in the length of hospital stay in CAP can have a significant cost impact, and even a one-day reduction in length of stay has been shown to yield substantial cost savings [15,24]. Our findings are compatible with the results of a prospective observational cohort study in Israel which demonstrated that a greater decrease in CRP levels between the first and second day of hospitalization was associated with a shorter length of hospital stay [25]. Similarly, a failure of CRP to decline by day 3 of hospitalization has been associated with prolonged hospital stay [13].

There are limitations to this study. First, the decision to admit, as well as to discharge, the patient with CAP was made by the clinician and was not based on specific, pre-defined criteria. However, almost 80% of the admitted patients had a CURB-65 score ≥2 and fewer than 4% had a CURB-65 score of 0, indicating that the more severe patients were hospitalized. The lack of data regarding comorbidities and past vaccinations that are known to influence susceptibility to CAP is a major limitation of our study. Third, treating physicians—although in the same department—may have started antibiotic treatment with different regimens, as there was no specific protocol for guiding antibiotic therapy, and therefore, ΔCRP could not be strongly associated with successful treatment. However, the very low death rate demonstrates that, in the majority of CAP patients, the chosen antibiotic regimen was successful, but it may also be attributed to the lower mean age of our study group. Another limitation is the low number of patients with an established microbiological diagnosis. However, this is common in clinical practice. Moreover, since D1 was defined as the day of hospital admission and initiation of empirical antibiotic treatment, the time between the onset of patients’ symptoms and hospital admission varied among patients and may have influenced the absolute CRP values at D1, and that is a limitation. Eventually, the study was conducted in a single center, and accordingly, this makes the generalizability of our findings ambiguous. A strength of this study is the relatively high number of patients, without any lost to follow-up, and the fact that it reflects common clinical practice, as in a real-life situation. None of the included patients had received antibiotic treatment prior to hospitalization, and thus, they constituted a homogenous population of in-patients with CAP.

Undoubtedly, there is need for further research on how to use the information provided by single biomarker measurements and to corroborate the additive value of these biomarkers in order to improve clinical decision-making regarding the management and prognosis of hospitalized patients with CAP in daily practice. In conclusion, we found that CRP levels on D4 and D7 could fairly predict CAP survival, and a reduction of CRP >50% by D4 of hospitalization corresponded to four fewer days of length of hospital stay.

## Figures and Tables

**Figure 1 jpm-12-01710-f001:**
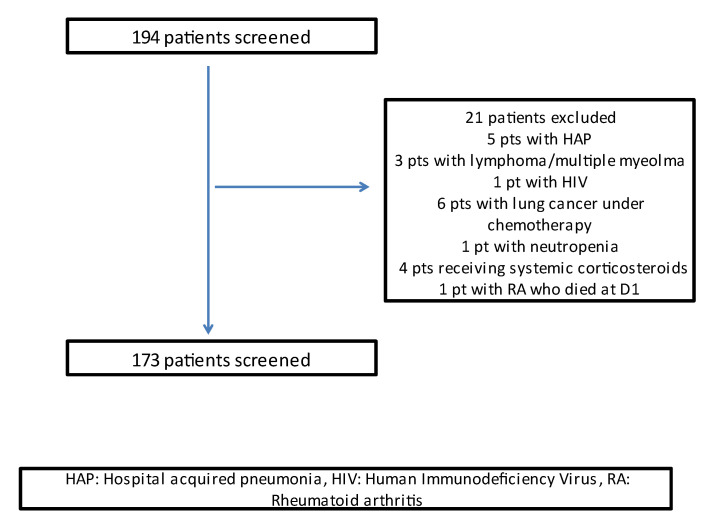
Flowchart of the study participants.

**Figure 2 jpm-12-01710-f002:**
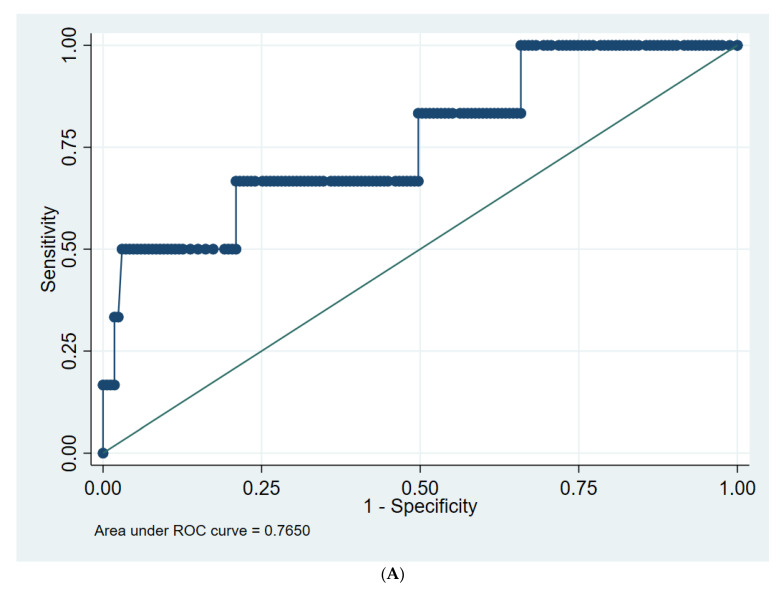
(**A**) ROC curve for CRP D4 in predicting CAP survival; (**B**) ROC curve for CRP D7 in predicting CAP survival.

**Table 1 jpm-12-01710-t001:** Characteristics of patients with CAP.

Subjects (*n* = 173)	Survivors (*n* = 167)	Non-Survivors (*n* = 6)	*p*-Value
Age	62.6 ± 21.4	82.8 ± 13.7	0.011
Sex (M/F)	98/69	6/0	0.112
Smoking (C/Ex/N)	65/47/55	1/4/1	0.115
CRP D1	11.8 ± 9.1	17.8 ± 12.5	0.216
CRP D4	6.4 ± 6.2	16.9 ± 13.5	0.054
CRP D7	4.6 ± 5.8	12.8 ± 10.7	0.059
Length of stay (days)	14.4 ± 12.1	16.5 ± 9.2	0.397
pO2/FiO2	268.9 ± 76.2	210.5 ± 68.9	0.074
WBC	12,277 ± 4684	12,688 ± 5008	0.816
CURB-65 score			0.513
(0,1,2,3,4)	7/27/70/50/13	0/0/2/3/1	

Data are presented as *n*, mean ± SD. CRP: C-reactive protein, WBC: white blood cell count, CURB: confusion, urea, respiratory rate, and blood pressure.

## Data Availability

Not applicable.

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
