# Peer review of "C-Reactive Protein as a Predictor of Survival and Length of Hospital Stay in Community-Acquired Pneumonia"

_jpm, 2022, doi:10.3390/jpm12101710_

Round 1

Reviewer 1 Report

The present study is interesting and describes a fairly common situation of using CRP as an inflammatory biomarker with the aim to predict survival and length of hospital stay in patients hospitalized due to community acquired pneumonia.

However, the study findings demonstrate no association between CRP values on day 1, day 4 and day 7 and the ability to predict survival. The study lacks in depth insights and discussion of its implications. The authors do not provide any suggestion for the clinician regarding how to cope with this dilemma of treating CAP patients- When the clinician encounters a CAP patient with an elevated CRP level compared to another CAP patient with normal CRP level both at the same day of hospitalization- does the clinician need to act differently when treating these two patients?

No data regarding chronic illness of the participants is provided. Such information is crucial considering baseline susceptibility to CAP. What is the patient relevant past medical history? (How many had COPD and not just smokers? How many diabetic patients? How many had chronic renal disease? Etc.). How many received influenza vaccine? How many pneumococcal vaccine? COVID-19 vaccine?

Were all the patients hospitalized in the department of respiratory medicine? What are the characteristics of respiratory medicine department compared to the regular medicine department and the ICU?

Why were days 1, 4 and 7 of hospitalization chosen by the authors for CRP measurement and kinetic calculation? (In the introduction section of the manuscript the authors describe a study with 25% decrease in CRP on day 2. Furthermore, in the discussion section the authors describe a study from Denmark, which examined a decline of CRP on day 3). So why were these three specific days of hospitalization chosen?

Why were patients who died within the first two days of hospitalization excluded? It would be interesting to know their admission CRP levels compared to the patients who were included in the study.

I suggest adding a diagram that demonstrates inclusion and exclusion from the study: how many met the inclusion criteria, how many met the exclusion criteria and which exclusion criteria. This will better enable the reader to grasp the magnitude of the proposed study.

In addition to the sputum cultures, blood cultures and urine antigen, were the patients assessed for respiratory viral infection (influenza, COVID-19 etc.)?- Such information is important considering severity of infection and hence CRP values.

The authors describe the absolute CRP values on day 1, day 4 and day 7. However, what the authors describe as “kinetics” of CRP is not actually kinetic. It is merely a reduction of one CRP absolute value from another CRP absolute value without taking into account in the calculation the element of time between the tests. Even tough the authors describe that the “kinetics” calculation compared day 4 to day 7 etc., the element of time is not part of the mathematical calculation, which is absolutely necessary when describing kinetics (I refer the authors to the recently published review of the CRP velocity biomarker by Levinson et al in the International Journal of Molecular Science, which describes the use of kinetic properties of the CRP biomarker for further details).

The authors do not provide any information regarding the time difference between the appearance of patients’ symptoms compared to time they seek medical care. Such information is vital when counting on CRP to assess the severity of infection (For example: a patient suffering for 1 day of untreated pneumonia would be expected to have a different CRP level compared to a patient suffering from 4 days of untreated pneumonia).

The authors describe the differences of antibiotics prescribing.  However, the authors provide no data regarding the susceptibility of bacteria in the positive cultures with the specific antibiotic treatment the patient with the positive culture had received.

Among non-survivors, at what day of hospitalization did the 6 patients die?

Please compare the positive cultures between survivors and non-survivors.

It was shown in the past that a repeat CRP measurement might ameliorate the ability of the clinician to better identify the developing inflammatory process associated with infection. Furthermore, recent studies have shown the utility of using CRP’s kinetic properties for this purpose. In the presented study the absolute CRP values and the so-called kinetic CRP values did not manifest this association with the developing inflammatory process- there should be some discussion regarding these differences, their cause and meaning.

Minor comments:

There is a bracket missing in the last line of the “CAP survival” subsection of the results section. Please correct.

Punctuation signs should be placed in the correct place throughout the manuscript.

Author Response

REVIEWER 1

The present study is interesting and describes a fairly common situation of using CRP as an inflammatory biomarker with the aim to predict survival and length of hospital stay in patients hospitalized due to community acquired pneumonia.

However, the study findings demonstrate no association between CRP values on day 1, day 4 and day 7 and the ability to predict survival. The study lacks in depth insights and discussion of its implications. The authors do not provide any suggestion for the clinician regarding how to cope with this dilemma of treating CAP patients- When the clinician encounters a CAP patient with an elevated CRP level compared to another CAP patient with normal CRP level both at the same day of hospitalization- does the clinician need to act differently when treating these two patients?

Response: The reviewer has a point. Obviously, a normal or slightly elevated CRP level may indicate a less severe CAP. However, according to guidelines, the empirical treatment in patients with CAP does not depend on CRP levels. A comment has been added in the discussion that reads as follows: “Currently, CRP and PCT are the best available tools to assess the severity of CAP”

No data regarding chronic illness of the participants is provided. Such information is crucial considering baseline susceptibility to CAP. What is the patient relevant past medical history? (How many had COPD and not just smokers? How many diabetic patients? How many had chronic renal disease? Etc.). How many received influenza vaccine? How many pneumococcal vaccine? COVID-19 vaccine?

Response: Our study aimed at assessing the predictive performance of absolute CRP values and kinetics in the survival and length of hospital stay regardless of past medical history and was a real life study. Accordingly, apart from patients with CAP that did not fulfill the inclusion and exclusion criteria all other patients were included in the study.

Were all the patients hospitalized in the department of respiratory medicine? What are the characteristics of respiratory medicine department compared to the regular medicine department and the ICU?

Response: All study patients were hospitalized in the 1st University Department of Respiratory

Medicine, University of Athens. In case of deterioration, and according to clinical judgment admittance to the ICU attached to our Department occurred. A comment has been added in the Results that reads as follows: “All of them were hospitalized in the 1st University Department of Respiratory Medicine, University of Athens, Greece.”

Why were days 1, 4 and 7 of hospitalization chosen by the authors for CRP measurement and kinetic calculation? (In the introduction section of the manuscript the authors describe a study with 25% decrease in CRP on day 2. Furthermore, in the discussion section the authors describe a study from Denmark, which examined a decline of CRP on day 3). So why were these three specific days of hospitalization chosen?

Response: The reviewer is right in that the selection of Days 1, 4 and 7 was arbitrary. The same applies to the choice of D2 in the other studies. Our clinical rationale for D4 is that according to guidelines an antibiotic regimen requires at least 3 days to have an effect and ideally should not change unless a culture result or significant deterioration occur. As for D7, it was chosen because most guidelines indicate a 5-7 day duration of antibiotic course. A comment has been added in the discussion  that reads as follows: “In the present study we aimed to evaluate the predictive performance of CRP in survival and hospital length of stay in hospitalized patients with CAP. The choice of D1, D4 and D7 was arbitrary and was based upon the clinical course of CAP, thus at admittance (D1), after 3 days of antibiotic treatment when re-evaluation usually occurs (D4) and towards the end of antibiotic treatment (D7).”

Why were patients who died within the first two days of hospitalization excluded? It would be interesting to know their admission CRP levels compared to the patients who were included in the study.

Response: For those patients who died within the first 2 days, no evaluation of kinetics could be performed. However, it should be mentioned that only 1 patient died at D1 and was excluded from the study because he was also immunocompromised. 

I suggest adding a diagram that demonstrates inclusion and exclusion from the study: how many met the inclusion criteria, how many met the exclusion criteria and which exclusion criteria. This will better enable the reader to grasp the magnitude of the proposed study.

Response: The diagram (flow chart) has been added according to the reviewer’s suggestion as Figure 1.

In addition to the sputum cultures, blood cultures and urine antigen, were the patients assessed for respiratory viral infection (influenza, COVID-19 etc.)?- Such information is important considering severity of infection and hence CRP values.

Response: The study was conducted before the COVID-19 pandemic. Moreover, no test for virus detection was performed as routine test. A comment for the time period of the study has been added in the Methods section.

The authors describe the absolute CRP values on day 1, day 4 and day 7. However, what the authors describe as “kinetics” of CRP is not actually kinetic. It is merely a reduction of one CRP absolute value from another CRP absolute value without taking into account in the calculation the element of time between the tests. Even tough the authors describe that the “kinetics” calculation compared day 4 to day 7 etc., the element of time is not part of the mathematical calculation, which is absolutely necessary when describing kinetics (I refer the authors to the recently published review of the CRP velocity biomarker by Levinson et al in the International Journal of Molecular Science, which describes the use of kinetic properties of the CRP biomarker for further details).

Response: We agree with the reviewer’s comment. However, we cannot assess the estimated CRP velocity since we do not have data regarding the time from symptom onset to hospital admission (and thus first CRP measurement). As for the serial measurements we have used the same time difference for all patients with D1, D4 and D7. As we describe in the Methods section – statistical analysis - the difference Δ was calculated by the formula: Δ = D4-D1, D7-D4 and D7-D1, respectively. Therefore, ΔCRP4-1 = CRPD4-CRPD1, ΔCRP7-4 = CRPD7-CRPD4, ΔCRP7-1 = CRPD7-CRPD1, where Δ>0 refers to increasing values and Δ≤0 refers to reducing values.  We have used the term “kinetics” for this instead of “dynamics”. The same term (kinetics) has been used by our group in a study evaluating CRP and PCT in Ventilator-associated pneumonia (Eur Respir J 2010; 35(4):805-811)

The authors do not provide any information regarding the time difference between the appearance of patients’ symptoms compared to time they seek medical care. Such information is vital when counting on CRP to assess the severity of infection (For example: a patient suffering for 1 day of untreated pneumonia would be expected to have a different CRP level compared to a patient suffering from 4 days of untreated pneumonia).

Response: Our study was conducted in a real life setting. The day of CAP diagnosis was defined as D1 and was the day that empirical antibiotic treatment was started regardless of symptom onset. Collected data, such as CURB-65 score and pO2/FiO2 ratio on D1 may reflect the severity of CAP and were assessed along with CRP (as mentioned in the Results a positive correlation was detected between PaO2/FiO2 and CRPD4 or CRPD7). Additionally, the so called “CRP kinetics” were evaluated to show the trend (increase or reduction) of the biomarker in order to assess the prognostic performance regardless of time from the onset of symptoms. A comment has been added in the limitations of the study that reads as follows: “Moreover, since D1 was defined as the day of hospital admission and initiation of empirical antibiotic treatment the time between the appearance of patients’ symptoms and hospital admission varied among patients and may influence absolute CRP values at D1.”

The authors describe the differences of antibiotics prescribing.  However, the authors provide no data regarding the susceptibility of bacteria in the positive cultures with the specific antibiotic treatment the patient with the positive culture had received.

Response: The number of patients with a positive blood or sputum culture was rather small. Antibiotic treatment was modified according to the susceptibility of the isolated pathogen in those patients with a positive culture in case empirical treatment did not cover the pathogen. A comment has been added in the Results that reads as follows: “Antibiotic treatment was chosen by the treating physician and was modified according to the susceptibility pattern of the sputum and/or blood culture in case empirical treatment did not cover the isolated pathogen.”

Among non-survivors, at what day of hospitalization did the 6 patients die?

Response: Among non-survivors, the 6 patients died on the 3rd, 8th, 14th, 17th, 21st and 23th day of hospitalization. A comment has been added in the Results that reads as follows: “One death occurred at D3 of hospitalization and 5 non-survivors died from D8 to D23.”

Please compare the positive cultures between survivors and non-survivors.

Response: Only one of non-survivors was found to have a positive blood culture. The other 5 non-survivors did not yield a pathogen in blood cultures. A comment has been added in the Results that read as follows: “One patient with a positive blood culture died while no pathogen was isolated in the cultures of the other 5 non-survivors.”  

It was shown in the past that a repeat CRP measurement might ameliorate the ability of the clinician to better identify the developing inflammatory process associated with infection. Furthermore, recent studies have shown the utility of using CRP’s kinetic properties for this purpose. In the presented study the absolute CRP values and the so-called kinetic CRP values did not manifest this association with the developing inflammatory process- there should be some discussion regarding these differences, their cause and meaning.

Response: The reviewer is right. In our study absolute CRP values and the so-called CRP kinetics did not manifest this association with the developing inflammatory process as shown in other studies. This discrepancy may be attributed to the low number of deaths in our study population. Only 6 out of 173 patients died corresponding to a 3.47% mortality which is lower compared to other studies. Moreover, the mean age of study participants and especially survivors was lower compared to other studies, possibly contributing to the low mortality. A comment has been added in the Discussion that read as follows: “This discrepancy may be attributed to the low number of deaths in our study population. Only 6 out of 173 patients died and this corresponds to a 3.47% percentage which is lower compared to other studies. Moreover, the mean age of survivors was lower compared to other studies, possibly contributing to the low mortality since mortality has been shown to increase with age.”

Minor comments:

There is a bracket missing in the last line of the “CAP survival” subsection of the results section. Please correct.

Response: The bracket has been added.

Punctuation signs should be placed in the correct place throughout the manuscript.

Response: Punctuation signs have been checked throughout the manuscript.

Reviewer 2 Report

Comments:

Page 3, paragraph 2: It could be added here that CRP is not only an inflammation marker, but also stimulates inflammation itself.

Literature e.g.: Kaplan, M.H.; Volanakis, J.E. Interaction of C-reactive protein complexes with the complement system. I. Consumption of human complement associated with the reaction of C-reactive protein with pneumococcal C-polysaccharide and with the choline phosphatides, lecithin and sphingomyelin. J. Immunol. 1974, 112, 2135–2147.

Page 4 3rd line: Proposed addition: The study was conducted at the 1st University Department of Respiratory Medicine, University of Athens.

Page 9 1st paragraph: 'Only 6 out of 173 patients died, a percentage of 3.47%. Percentage that is lower compared to other studies'. 

The average age of the survivors is 62.6 years, which is significantly younger than in many comparative studies, so that the average age should also be mentioned here.

Page 10 1st paragraph: In many studies, the survival rate in CAP correlates less with the administration of antibiotics than with the age of the patients. This should at least be mentioned here.

Author Response

REVIEWER 2

Page 3, paragraph 2: It could be added here that CRP is not only an inflammation marker, but also stimulates inflammation itself.

Literature e.g.: Kaplan, M.H.; Volanakis, J.E. Interaction of C-reactive protein complexes with the complement system. I. Consumption of human complement associated with the reaction of C-reactive protein with pneumococcal C-polysaccharide and with the choline phosphatides, lecithin and sphingomyelin. J. Immunol. 1974, 112, 2135–2147.

Response: We have added a comment and the respective reference along with another one.

Page 4 3rd line: Proposed addition: The study was conducted at the 1st University Department of Respiratory Medicine, University of Athens.

Response: The proposed sentence has been added in the Methods.

Page 9 1st paragraph: 'Only 6 out of 173 patients died, a percentage of 3.47%. Percentage that is lower compared to other studies'. 

The average age of the survivors is 62.6 years, which is significantly younger than in many comparative studies, so that the average age should also be mentioned here.

Response: The reviewer is right. A comment, along with a reference, has been added that reads as follows: “Moreover, the mean age of survivors was lower compared to other studies, possibly contributing to the low mortality since mortality has been shown to increase with age”

Page 10 1st paragraph: In many studies, the survival rate in CAP correlates less with the administration of antibiotics than with the age of the patients. This should at least be mentioned here.

Response: The reviewer is right. A comment has been added to emphasize the importance of age in the mortality of CAP.

Round 2

Reviewer 1 Report

The authors have addressed many of my previous comments, however some remained unanswered or the answers are not satisfying.

As response to my comment regarding lack of data regarding chronic illness among the participants and past vaccinations that effect their susceptibility to infection, the authors replied that the "study aimed at assessing the predictive performance if CRP values…regardless of past medical history and was a real life study".

This reply is not satisfying and does not attempt to address my comment. Past medical history IS PART OF REAL LIFE and effects patients' presentation and patients' CRP level at presentation. At the very least, if authors are unable to provide satisfying data regarding past medical history and past vaccination status of the patients, this should appear as a major limitation of this study.

I commented about the usage of the terms "kinetics" and the authors respond that "we do not have data regarding the time from symptoms onset to hospital admission"- Again this is highly important when considering the meaning of elevated CRP at presentation and must be added as a limitation of the study.

I appreciate that the terms ”kinetics" was used by the authors group in a previous publication (EUR Respir J), however it DOES NOT change the fact the term "kinetics" doesn't apply here, rather a more suitable term is "Δ CRP". Please rephrase the definition accordingly throughout the manuscript.

Author Response

The authors have addressed many of my previous comments, however some remained unanswered or the answers are not satisfying.

As response to my comment regarding lack of data regarding chronic illness among the participants and past vaccinations that effect their susceptibility to infection, the authors replied that the "study aimed at assessing the predictive performance if CRP values…regardless of past medical history and was a real life study".

This reply is not satisfying and does not attempt to address my comment. Past medical history IS PART OF REAL LIFE and effects patients' presentation and patients' CRP level at presentation. At the very least, if authors are unable to provide satisfying data regarding past medical history and past vaccination status of the patients, this should appear as a major limitation of this study.

Response: The reviewer is right. The lack of data regarding comorbidities and past vaccinations that both affect the susceptibility to pneumonia is a limitation. A comment has been added in the discussion to address this limitation that reads as follows: “The lack of data regarding comorbidities and past vaccinations that are known to influence susceptibility to CAP is a major limitation of our study”.

I commented about the usage of the terms "kinetics" and the authors respond that "we do not have data regarding the time from symptoms onset to hospital admission"- Again this is highly important when considering the meaning of elevated CRP at presentation and must be added as a limitation of the study.

Response: We agree with the reviewer that the time period between onset of symptoms and admission to the hospital affects absolute CRP levels at D1. A comment has been added to address this point that reads as follows: “Moreover, since D1 was defined as the day of hospital admission and initiation of empirical antibiotic treatment the time between the onset of patients’ symptoms and hospital admission varied among patients and may influence absolute CRP values at D1 and that is a limitation.”

I appreciate that the terms ”kinetics" was used by the authors group in a previous publication (EUR Respir J), however it DOES NOT change the fact the term "kinetics" doesn't apply here, rather a more suitable term is "Δ CRP". Please rephrase the definition accordingly throughout the manuscript.

Response: The term “kinetics” has been replaced by ΔCRP throughout the manuscript.